# Evaluating possible intended and unintended consequences of the implementation of alcohol minimum unit pricing (MUP) in Scotland: a natural experiment protocol

Srinivasa Vittal Katikireddi,[1] Clare Beeston,[2] Andrew Millard,[1] Ross Forsyth,[1] Paolo Deluca,[3] Colin Drummond,[4] Douglas Eadie,[5] Lesley Graham,[6] Shona Hilton,[1] Anne Ludbrook,[7] Gerry McCartney,[2] Thomas Phillips,[8] Martine Stead,[9] Allison Ford,[10] Lyndal Bond,[11] Alastair H Leyland[1]

For numbered affiliations see end of article.

**Correspondence to**
Dr Andrew Millard;
Andrew.millard@glasgow.ac.uk

## ABSTRACT

**Introduction** Scotland is the first country to carry out a national implementation of minimum unit pricing (MUP) for alcohol. MUP aims to reduce alcohol-related harms, which are high in Scotland compared with Western Europe, and to improve health equalities. MUP is a minimum retail price per unit of alcohol. That approach targets high-risk alcohol users. This work is key to a wider evaluation that will determine whether MUP continues. There are three study components.

**Methods and analysis** Component 1 sampled an estimated 2800 interviewees at a baseline and each of two follow-ups from four Emergency Departments in Scotland and Northern England. Research nurses administered a standardised survey to assess alcohol consumption and the proportion of attendances that were alcohol-related. Component 2 covered six Sexual Health Clinics with similar timings and country allocation. A self-completion survey gathered information on potential unintended effects of MUP on alcohol source and drug use. Using a natural experiment design and repeated cross-sectional audit, difference between Scotland (intervention) and North England (control) will be tested for outcomes using regression adjusting for differences at baseline. Differential impacts by age, gender and socioeconomic position will be investigated. Component 3 used focus groups with young people and heavy drinkers and interviews with stakeholders before and after MUP implementation. The focus groups will allow exploration of attitudes, experiences and behaviours and the potential mechanisms by which impacts arise. The interviews will help characterise the implementation process.

**Ethics and dissemination** Study components 1 and 2 have been ethically approved by the NHS, and component 3 by the University of Stirling. Dissemination plans include peer-reviewed journal articles, presentations, policy maker briefings and, in view of high public interest and the high political profile of this flagship policy, communication with the public via media engagement and plain language summaries.

**Trial registration number** ISRCTN16039407; Pre-results.

### Strengths and limitations of this study

► This evaluation adopts multiple methods to help establish whether minimum unit pricing (MUP) has caused changes in alcohol-related attendances using a natural experiment design, which is the most appropriate for this topic as a randomised controlled trial would not be feasible.

► Our study exploits divergences in Scottish and English alcohol policy to evaluate the effectiveness of an all-beverage MUP for the first time, and evaluates both positive and possible negative impacts of MUP of alcohol; such negative impacts might include use of other sources of alcohol, other substances or reduction in money available for essentials.

► The Northern English control group is likely to be comparable to Scotland because of geographical proximity and similar levels of deprivation, but we also assess the external validity of our sample by reference to routine data on attendances at Emergency Departments and Sexual Health Clinics.

► There is the potential to follow-up individuals through longer term data linkage, for which we will obtain respondents' permission, thus adding a cohort dimension to the initial cross-sectional approach.

► The main limitation is that the non-randomised design risks selection bias, for example in the differential selection of intoxicated attendees for interview.

## INTRODUCTION

Minimum unit pricing (MUP) of alcohol is a novel public health policy that aims to reduce alcohol-related harms across the population. In May 2012, Scotland became the first country to pass legislation to introduce MUP without reference to beverage type, a politically high profile measure,[1–3] although some Canadian provinces have introduced beverage-specific MUP, as further described below.

This paper describes the protocol for three associated studies that collect primary data to evaluate the impact of the policy in Scotland. After delays following legal challenges,[4 5] MUP legislation was implemented in May 2018 but it is due to cease in 6 years unless shown to be effective, under a 'sunset clause'. These studies are a key part of the evaluation planned by NHS Health Scotland, which will report to Scottish Ministers and the Scottish Parliament. The evaluation includes analyses by others of routinely collected administrative data.[6]

The overall aim of the study is to investigate the impacts of MUP on acute and chronic health harms (including by deprivation and gender and age subgroups). The study will also determine the extent to which specific unintended consequences occur. The evidence will be a key input to the decision whether the policy will continue.

Alcohol consumption is a major determinant of population health globally,[7] and the level of alcohol-related harm in Scotland is high compared with the rest of Western Europe.[8] Alcohol is associated with over 200 medical conditions including an increased risk of liver disease, heart disease, unintended pregnancy, sexually transmitted infections, some cancers and some injuries.[9–14] Its impact extends beyond the individual, with adverse effects on families, communities and the wider economy.[15 16] Furthermore, alcohol is known to be a major contributor to socioeconomic inequalities in health in the UK and elsewhere.[17–20] From a public health perspective, alcohol therefore has considerable potential as a modifiable risk factor to be addressed to help reduce health inequalities.[21–23]

A consistent body of international evidence demonstrates a negative relationship between price, consumption and alcohol-related harms.[24–28] The WHO has identified changing the price of alcohol, and controls on promotion and availability, as key methods for addressing alcohol-related health harms.[29–31] Across much of the world, the best established mechanism for influencing price has been taxation through alcohol duty, although in many countries, including the UK, the primary purpose is revenue-raising for central government rather than achieving public health goals.[32] As a result of this secondary consideration of public health, alcohol duty may be poorly designed to address alcohol-related harms since duty typically varies by product in a manner not consistent with the associated harms arising from the product's consumption. At the time of writing tax on a litre of 7.5% alcohol by volume (ABV) cider is 40 pence, on a litre of wine of the same strength tax would be 289 pence.[33]

A potentially complementary approach to increasing alcohol duty is the introduction of a floor price, such as MUP, below which alcohol should not be retailed.[34 35] MUP sets a minimum price per UK unit (8 gm) of pure alcohol. Epidemiological studies have found that drinkers at the greatest risk of harm tend to consume the cheapest (per unit) alcohol, thereby providing evidence that MUP may better target harmful drinkers in comparison to alcohol duty.[36 37] Econometric modelling studies suggest that MUP will result in a greater reduction in health harms compared with an equivalent rise in taxation under the UK's current system of calculating alcohol duty.[21 22 38–40] In addition, setting a floor price minimises the potential for 'trading down' to cheaper drinks, given that alcoholic drinks below the floor price would no longer be legally available. This is particularly relevant since previously retailers have opted not to pass on alcohol duty increases to consumers, as indicated by the existence of below-cost products.[41]

Comparable interventions to MUP have been introduced elsewhere. The best known example is in Canada, where some Canadian provinces introduced a minimum price for selling specific beverages (also referred to as 'reference pricing') within the context of government-operated monopolies.[42 43] While some authors refer to these interventions as 'minimum pricing', the nature of this policy differs in some aspects from 'MUP' as planned in Scotland.[3 5 44] In contrast to MUP which applies a uniform minimum price per unit across all beverage types, reference pricing imposes differing minimum prices that are determined by both alcohol strength and drink type. In addition, MUP in Scotland was introduced into a competitive alcohol market at a national level, in contrast to the locally applied government-owned monopolies in which reference pricing has been introduced. Important benefits of reference pricing in Canada have been demonstrated, with reductions in alcohol consumption, alcohol-attributable hospital admissions and crime observed.[45–50] However, a study focusing on Emergency Department (ED) visits found no reduction in overall visits for alcohol-related injuries, although alcohol-related motor vehicle injuries did fall.[51] Other broadly comparable pricing interventions, such as the abolition of cheap vodka within communist Russia, have similarly been associated with reductions in alcohol-related mortality.[52]

In Scotland, qualitative research has investigated the policy process through which MUP developed, including assessing the role of commercial interests, and seeking to identify transferable lessons for public health advocacy.[3 5 44 53–57] The influence of econometric modelling has been specifically investigated.[58 59] The dominant media discourses and the roles of different policy stakeholders in articulating arguments to the public have been explored using content analysis of newspaper reporting and trends in newspaper coverage have been tracked over time.[60–63] The views of the public and heavy drinkers around MUP have also been investigated.[64–67] There remained a need to investigate the views of policy stakeholders, the public and heavy drinkers about MUP as implemented.

Empirical research gathering primary data specifically focused on MUP in Scotland remains limited. The introduction of MUP provides a unique opportunity to use a natural experiment to evaluate public health policy, and provide new real-world evidence on the effectiveness of MUP, where most evidence to date has been from modelling studies.[68 69] A broad programme of research is being co-ordinated by NHS Health Scotland, which

will report to Scottish Ministers and the Scottish Parliament between five and 6 years after the start of the policy. This programme includes analyses of administrative data and alcohol sales data.[70] The protocol described here complements the NHS-led work and has been funded by the National Institute for Health Research Public Health Research programme to collect primary data in three associated studies.

Much of the burden of alcohol-related harm, especially among young people, arises from acute harm following high episodic consumption ('binge drinking'). These harms are of particular interest to policymakers as they may be associated with social disorder and violence. That in turn provides one justification for MUP, although the case for MUP legislation in Scotland was fought (and won) on public health benefits. Comprehensive diagnostic data on ED alcohol-related attendances that do not result in admission are currently not adequately captured in administrative health data within the UK, so require primary research.

Our work focuses on both intended and possible unintended consequences of the intervention.[71] A number of potential risks arising from the introduction of MUP have been identified by policymakers, the alcohol industry and public health professionals.[54]

1. Consumers may switch to alternative sources of alcohol not subject to MUP so that the price paid does not increase. Such sources include both legal sources: internet sales from outside Scotland, legitimate cross-border purchase for own use and home fermentation and illegal sources (counterfeit and unlicensed sale of smuggled or stolen alcohol).
2. Increased alcohol-related harm could occur through substitution (eg, to illicitly produced or industrial alcohol associated with greater toxicity) or changed drinking patterns (eg, moving from regular drinking to binge drinking).
3. Displacement effects with reductions in alcohol-related harms potentially accompanied by increases in harms related to other substance use.
4. MUP could unfairly penalise deprived populations less able to absorb the additional financial cost and this may adversely affect access to essentials such as food and fuel.
5. MUP may have unintended effects on harmful drinkers who cannot reduce their consumption.
6. There may be adverse economic impacts on the Scottish alcohol industry retailers and/or manufacturers.

The ED and Sexual Health Clinics (SHC) studies described in this protocol will address the first four of these potential risks, although analysis of existing datasets by NHS Health Scotland will further address the potential impact on access to essential goods. A study has been commissioned from the University of Sheffield by NHS Health Scotland to assess the effects on harmful drinkers. The possible adverse economic impact on the Scottish alcohol industry will not be addressed by our project but will be monitored by the Scottish Government through the Monitoring and Evaluating Scotland's Alcohol Strategy (MESAS) studies managed by NHS Health Scotland.

It was not possible to collect sufficiently valid consumption data in the ED setting to assess alcohol consumption levels based on consumption owing to interview time constraints. Instead, we defined hazardous, harmful and dependent alcohol consumption using scores from the Fast Alcohol Screening Test (FAST).[72] The use of FAST to identify alcohol misuse (ie, hazardous, harmful and dependent alcohol consumption) has been supported by national guidelines for the prevention of harmful drinking.[73] Derived from the AUDIT[74] the four item FAST was specifically developed and validated within ED as it is quick to administer and less disruptive in the busy ED setting.[72 75] The FAST has been found to be the most sensitive and accurate short alcohol screening tool commonly administered to identify alcohol misuse in adults attending ED.[76 77] and has been used in a number of clinical studies.[78 79] A score of ≥3 on the FAST indicates hazardous drinking. We are undertaking a parallel examination of The FAST screening tool to further discriminate alcohol use disorders in ED attendees. In our study MUP is theorised to reduce alcohol misuse (hazardous, harmful and dependent alcohol consumption), so reducing both acute and chronic alcohol-related harm, and therefore the numbers of alcohol-related attendances at both EDs and at SHCs. Hazardous drinkers are expected to be more likely than harmful or dependent drinkers to present with acute harms (such as injuries and assaults), while harmful/dependent drinkers are likely to present with both acute conditions and acute consequences of chronic conditions (such as pancreatitis and gastritis). Therefore there may be an early impact on ED attendances relating to chronic as well as acute alcohol-related conditions.

## RESEARCH QUESTIONS

The more specific research questions and objectives are summarised in table 1, with the contribution of each of the three study components stated.

Another separately funded study is focusing on the impacts on people with alcohol dependence, including high rates of homelessness specifically. We therefore do not target this group for our qualitative work.

At the time of revising this paper for referees' comments, all data have been collected for all three components and analysis will shortly commence.

## METHODS AND ANALYSIS
### Component 1: audit of alcohol-related attendances at EDs
#### Study design

This component is designed to quantify changes in alcohol-related attendances at ED settings as a result of MUP. The FAST is used to identify alcohol misuse in the ED. The component focuses on answering research aim (RA) 1, namely:

 

**Table 1** Research aims and study components

| Research aims (RAs) | Study components (C) |
|---|---|
| RA1: To determine the impact of MUP on alcohol-related harms and drinking patterns for the overall population and by subgroups of interest (age, sex and socioeconomic position). | ED study of alcohol-related attendances (C1); Survey of alcohol-related behaviours (consumption patterns, alcohol spend, source of alcohol, move to other substances) in SHCs (C2) |
| RA2: To determine the impact of MUP on non-alcohol substance use, and other unintended impacts, for the overall population and by subgroups of interest (age, sex and socioeconomic position). | Survey of alcohol-related behaviours in SHCs (C2); Qualitative focus group study and stakeholders (C3) |
| RA3: To describe changes in experiences and norms towards MUP and alcohol use following the introduction of MUP by subgroups of interest (age, sex and socioeconomic position). | Qualitative focus groups with young people/heavy drinkers and interviews with stakeholders from public services (C3) |

ED, Emergency Department; MUP, minimum unit pricing; SHC, Sexual Health Clinics.

1. What are the impacts of MUP for alcohol on alcohol-related harms and drinking patterns (using the FAST) for ED attendees and by subgroups of interest (age, sex and deprivation)?
2. Does the effect of MUP vary dependent on the type of alcohol-related harm:
   a. Acute alcohol-related harms vs chronic alcohol-related harms?
   b. Broad diagnostic groups (based on coding systems used in EDs?
3. Does MUP affect alcohol consumption (on the basis of the FAST[72] score and alcohol misuse over the reporting period among people attending EDs (FAST ≥3)?
4. Does the MUP intervention effect size (assessed on a variety of measures including FAST) vary at the second and third time points?

Lower level outcomes are set out in the Data collection section. To answer RA1, a natural experiment design employing a repeated cross-sectional audit of all alcohol-related attendances at EDs in two Scottish hospitals and two North of England comparator hospitals at three time points was most appropriate to assess causality. A Randomised Controlled Trial was not feasible. The study methods and tools were based on previous studies used to quantify the national prevalence of alcohol-related attendances in EDs in England[79] and informed by the experience of the Scottish ED Alcohol Audit, carried out across 15–20 hospitals throughout mainland Scotland between October 2005 and June 2007.[80]

### Study population
People aged 16 years and over attending EDs during data collection periods were approached. The study monitored, but excluded from interview:
► those acutely physically or mentally ill such that informed consent or full participation was impossible to obtain (including gross intoxication),
► those with whom it was impossible to adequately communicate (eg, patients unable to speak English despite the use of an interpreter where available), and

► those who left the department before an approach was made or who were deemed by the research nurses as too threatening to approach.

Reasons for not approaching were recorded to address potential selection bias.

### Data collection
Baseline data collection took place in February 2018 prior to the implementation of MUP on 1st May 2018. The second and third waves of data collection took place approximately 8 and 12 months post baseline and 5 and 9 months post implementation, with care taken to avoid the inclusion of specific events that are expected to be associated with increased or decreased alcohol consumption, for example local holidays were checked to avoid differences between groups. Each wave collected data over 3 weeks, and to maximise collection of both alcohol-related attendances and alcohol misuse, data collection occurred from 20:00 to 03:30 from Thursday to Sunday (ie, 30 hours per week during this time period) and 09:00 to 16:30 from Monday to Wednesday (ie, 22.5 hours per week during this time period).

During study data collection periods, patients were approached by a research nurse to seek permission to complete a short face-to-face interview. All eligible patients were approached, informed consent obtained and a face-to-face interview based on a previously used patient questionnaire was administered electronically using a tablet (iPad).[78] An ED attendance database was maintained for all attendances during data collection periods to allow monitoring of patient recruitment.

The interview collected details on the following:
► basic demographic information (age; gender; ethnicity; marital, employment and housing status (including unstable housing status such as no fixed abode and hostel accommodation); postcode sector);
► attendance details (date and time, triage category, time and date of incident);
► how much alcohol was consumed in the past 24 hours;
► date, time and place of last drink;

► the largest number of drinks on any 1 day in the previous week;

► self-reported reason for attendance, such as assault, collapse etc;

► alcohol misuse in the past year (as assessed by the FAST)[72] and

► whether alcohol consumption changed over the previous 12 months.

In addition we asked whether interviewees thought their current ED attendance was related to their or someone else's drinking, and how many times they had attended any ED in the previous 12 months.

A note was made of the hospital unique identifier for the attendance if participants agreed to link their research interview data to their ED data. Research staff retrieved additional data from patient records on full postcode, triage category, discharge status and diagnoses at discharge (coded by International Classification of Diseases (ICD)-10 or other ED coding systems in use locally). The consent procedure asked participants for separate permissions both for this and for data linkage to facilitate longitudinal follow-up. Additionally, aggregated data on age group, sex and diagnosis were sought for all attendees in the study time periods in order to allow an assessment of the representativeness of the interviewee sample.

### Socioeconomic position and inequalities

Analysis by deprivation, age and sex will be undertaken to determine if the impact of MUP on alcohol-related attendances at EDs is differentially patterned. Assessment of alcohol-related attendances within the ED setting provides a broad range of deprivation, age, sex and drinking status categories. It therefore not only provides information on an important primary health outcome but also allows adequate investigation of the intervention's impact on important subgroups.

### Outcome measures

The primary outcome will be the proportion of attendances that are alcohol-related, using a composite measure which defined the attendance as alcohol-related if any one (or more) of the following was true:

► patient self-reports attendance is alcohol-related;

► patient reports alcohol consumption in past 24 hours ≥8 units in men, ≥6 units in women; or

► patient not approached or interview terminated because too intoxicated with alcohol.

Secondary outcomes will be:

► the absolute number and proportion of alcohol-related attendances;

► alcohol misuse defined by the FAST score ≥3;

► mean FAST score by age, gender and deprivation;

► prevalence of binge drinking in the past week, and

► self-reported reason for attendance.

Outcomes will be compared between intervention and control areas, adjusted for baseline attendances.

### Power calculation

Based on the experience of the 24 hours audit of EDs in England,[79] and assuming that we would be able to recruit 50% of eligible ED attendees, we anticipated that the four sites (two Scotland, two England) would result in 940 recruits per week. Recruiting over 3 weeks pre-implementation and post-implementation—giving a total sample size of 5640 for the baseline and any one subsequent wave—would mean that we would be highly powered (>80%) to detect an effect size of ±5% in the proportion of alcohol-related attendances from an estimated 30% with 95% significance. We have used a base rate of 30% informed by the 24 hours audit of EDs in England[79] and assumed a 5% decrease would be of public health importance and may be expected based on current evidence.[79] For subgroup analyses, we would have good power (>80%) to detect an effect size of 0.23 on the FAST score among those from the most deprived quintile (estimated to be 25% of attendances) and an effect size of 0.27 among those aged 18–24 years (estimated to be 15% of attendances).

### Statistical analysis

We will test for differences in the outcomes between the intervention and control groups using a regression model with ED included as a fixed effect, and with individuals nested within the ED of their attendance, before and after adjustment for relevant covariates including baseline levels of alcohol-related attendance, age, gender, deprivation quintile and disease diagnostic categories. We will also attempt to determine the nature of the effect more precisely in terms of whether there is a dose-response effect according to the time since MUP was implemented (through a test of the significance of an interaction between time and intervention). We will test for interactions of the intervention with defined important covariates (including baseline levels of alcohol-related attendance, age, gender, deprivation quintile and disease diagnostic categories) to investigate the possibility of differential intervention effects and will subsequently stratify the analyses if indicated.

### Ethical arrangements

Trained nurse researchers approached and interviewed participants in order to ensure informed consent was obtained from study participants. The research team has considerable expertise with appropriately considering ethical issues in studies with similar methodology previously.[79 80] We followed standard procedures for obtaining informed consent. Participants received information on local alcohol services and national advice telephone lines in the participant information leaflet.

### Component 2: survey of alcohol-related behaviours in sexual health services

Young adults and high-risk alcohol users are under-represented in national surveys. SHCs were selected as the setting for data collection to provide adequate inclusion

of deprived groups and young people. Understanding how MUP might change alcohol misuse patterns in this group and/or increase use of other substances is considered a priority by policymakers. SHCs provide a convenience-sampling frame to monitor changes in behaviour within a high-risk population that includes adequate representation of both deprivation and young age. This component contributes to RA1 and RA2, specifically to investigate, among a population at high-risk of alcohol and drug-related problems:

1. changes in alcohol misuse,
2. how sources of alcohol consumption have changed following MUP,
3. whether MUP has impacted on the use of psychoactive substances apart from alcohol,
4. whether any unintended impacts of MUP, such as changes to alcohol sources and psychoactive substance use differ across age group, gender, highest educational attainment and employment status,
5. whether any observed intervention effects vary at the second and third time points.

### Study design
A repeated cross-sectional survey of patients attending SHCs was conducted to determine changes in drinking patterns and psychoactive substance use within this population. All patients attending three SHCs in Scotland and three in North England were invited to self-complete a short questionnaire. Data collection took place over 3 weeks at baseline and 8 and 12 months post-baseline. Timing was similar to the ED audit.

### Study population
Patients of any age attending sexual health services participating in the study during data collection periods were approached. Patients who are unable to understand English well enough to complete the questionnaire (with assistance), or who left the clinic before an approach was made, and those deemed inappropriate by clinical staff to approach were excluded from the study.

### Data collection
The questionnaire was offered to attendees at reception by a trained reception staff. Participants completed the questionnaire while waiting to be seen and returned it into a confidential returns box. The trained staff were available to offer assistance where needed. The confidential and voluntary nature of the study was emphasised. Similar patient completed questionnaires have been carried out as part of NHS Board work within NHS Lothian and NHS Greater Glasgow and Clyde achieving response rates of between 50% and 70% (without the assistance of support staff).

Questions covered where alcohol was procured, other substances used, the FAST tool (allowing quantification of alcohol misuse and binge drinkers) and basic characteristics (age band, sex, highest educational level). The questionnaire was designed to allow completion within between 2 and 5 min.

### Socioeconomic position and inequalities
Trained reception staff assisted self-completion if required where literacy problems or visual impairment otherwise precluded participation. Analysis will be undertaken by highest education level, age and sex to determine if behavioural response to MUP in terms of drinking, source of alcohol and displacement is differentially patterned.

### Outcome measures
The primary outcome from component 2, which aims to reveal unintended effects, will be the proportion of patients self-reporting psychoactive substance use other than alcohol in the last month. Secondary outcomes will be drug-specific rates of use within the last month, sources of alcohol purchase (including cross-border, internet, smuggled or moonshine), rates of higher risk drinking (as measured by FAST score>3), self-reported binge-drinking and analyses for differential displacement of use of illicit psychoactive drugs (including age, gender, highest achieved education, problematic drinkers)

### Power calculation
We envisaged being able to recruit at least 50% of attendees at sexual health clinics at the six sites (three Scotland, three England), therefore resulting in a total of slightly over 10 000 recruits over two 3 week periods (an average of 288 per clinic per week). This would give us power of more than 80% to detect a change of ±4% from an estimated 30% in the proportion of people using drugs.

### Statistical analysis
Similar analyses to the ED component will be undertaken based on regression models with site included as a fixed effect, and the proportion taking illicit psychoactive drugs as the dependent variable for the primary outcome. We will examine whether any displacement effects are sustained between six and 12 months post implementation. We will investigate interactions with defined important covariates (including age, gender, highest achieved education, problematic drinkers) to investigate the possibility of differential intervention harms.

### Ethical arrangements
To protect patient confidentiality, no identifiable personal data was collected. A research assistant was available to provide a questionnaire, answer any questions and assist those experiencing difficulty with self-completion. Given that young people are a particular focus of this study, no lower age restriction was planned. No separate written consent was planned as completion of the questionnaire was considered to constitute implied consent.

### Component 3: qualitative study of young people and heavy drinkers
The aim of this study component is to provide an in-depth understanding of how MUP is affecting key subgroups within the Scottish population. Understanding the experience of those exposed to the intervention allowed further

exploration of potential mechanisms that result in unanticipated benefits and harms and which may affect groups to different degrees. This study contributes specifically to RAs 2 and 3 by exploring participants' expectations and experiences of the policy's impact including unintended consequences, both personally and on family, friends and wider community, and by comparing narratives between the different sample groups: age, gender and socioeconomic position.

## Research design and data collection

Focus groups and interviews were carried out in January to April 2018 and repeated in September/October 2018 in three communities in Scotland; an affluent urban community and two deprived urban communities. All data were collected by a mixed gender team of four full time researchers at the University of Stirling with extensive experience of qualitative methods. All participants were residents or professional stakeholders located within the catchment areas served by the two Scottish hospitals participating in the ED study component. Postcode deprivation (Carstairs) scores were used to define study communities in each area which matched the socioeconomic comparators of interest (deprived and affluent). Those who expressed an interest in the study were provided a copy of the study information sheet and an opportunity to ask questions before consenting to take part. Discussion and face-to-face interviews were conducted in local community venues or stakeholders place of work in private spaces to maintain confidentiality. Separate interview schedules and topic guides were devised for each study group and revised between waves.

1. Focus groups were conducted with those subgroups who may be particularly affected by MUP: young people and heavy drinkers (previously categorised using the Alcohol Use Disorders Identification Test (version C) (AUDIT-C) tool, a brief three-item alcohol screening tool designed to identify hazardous and harmful drinking patterns) living in both deprived and affluent communities. Group participants were recruited by independent market researchers using door-to-door and street intercept techniques and were not known to the researchers. Recruitment was facilitated by the use of a recruitment questionnaire incorporating AUDIT-C screening questions. Each discussion covered social norms and attitudes, alcohol displacement behaviours and changing patterns in drinking and purchasing habits.

2. Semi-structured interviews with professional stakeholders positioned to observe immediate social, health and economic impacts of MUP were conducted at each wave. These participants were not known by the study team prior to the study commencing. As well as addressing the main research objectives, data from these interviews were used to characterise the implementation process within each study community from differing professional perspectives and to explore adequacy of implementation and any difficulties experienced. In addition, these data provided valuable contextual information for informing the focus group topic guides and identifying lines of enquiry specific to each study community.

## Study population

The focus groups recruited young and heavy drinkers (assessed as above) in three contrasting study communities: an affluent urban community and two deprived urban communities in Scotland. Focus group participants were recruited purposively (as described above), for the two population groups of interest, young binge drinkers aged 18–25 years and older heavy drinkers aged 30 years and over. At each wave 12 single-sex focus groups were planned with an average of five to six participants in each group. The number and composition of the groups were selected to enable comparisons across two key study dimensions: types of drinker and socioeconomic status; and to facilitate open and free discussion allowing each participant the time to express their views.

Semi-structured interviews were conducted at each wave with key stakeholders working within each of the three study communities. The professional groups from which the stakeholder sample was drawn included representatives from licensing bodies, police, youth services, general practice and alcohol and drug treatment services. Stakeholders were identified using cascading techniques with a ceiling of 15–18 interviews at each wave.

## Analysis plan

Focus group discussions and interviews were audio recorded and transcribed in full with participants' consent for analysis. Transcriptions were coded thematically using Nvivo12 software to allow the sharing of data across the research team and to enable identification of typical quotes, common reasoning and deviant or contradictory cases. Common themes and subthemes were identified across the two data sets (drinkers and stakeholders) to assess for continuity of findings and to provide a more vivid and rounded picture of community expectations, the implementation process and the legislation's short-term impact. Together it is anticipated that these data will help to illuminate the findings to emerge from other study components.

## PATIENT AND PUBLIC INVOLVEMENT

The development of the research question and outcome measures was informed by patients' priorities, experiences and preferences in two ways: through a review of media representations of MUP as a proxy source of information,[62] survey research on patient perceptions around alcohol consumption and price[81] and focus group and other research on public attitudes to MUP.[64–67]

Patients were not directly involved in the design of the study, but were interviewees and respondents and focus groups included members of the public. Service users with lived experiences of alcohol-related harms were

represented on the study steering committee. Results will be disseminated to the public by actively engaging the mass media, using plain language summaries, a website and posters at sites.

## DISSEMINATION PLAN

Given the policy relevance of this work, dissemination will be targeted to academic, policy and public audiences using tailored outputs.

Academic papers will include:

1. EDs study: the impact of MUP on alcohol-related attendances at EDs.
2. SHCs study: changes in intoxicating substance use among young people attending SHCs following MUP.
3. Qualitative study: public expectations and understandings of MUP and changes in social experiences of alcohol use following its introduction.

Further papers may be produced, dependent on the timing of data collection and following investigation for population subgroup-specific effects.

A series of short policy briefings and presentations will be produced to summarise both quantitative and qualitative findings. We will produce an overall theoretical interpretation using all three components (see table 1) in the final study report.

These will be primarily aimed at policymakers within the Scottish and UK governments, for example the Department of Health, Public Health England, Public Health Wales, the Public Health Agency in Northern Ireland and NHS Health Scotland, but will be disseminated more broadly, subject to demand. The work will also be included in the review report NHS Health Scotland will produce for the Scottish parliament and ministers. This will provide evidence for ministers' and the Scottish parliament's decision about whether to renew the MUP legislation or allow it to fall under the 'sunset clause'. The academic and health and social care policy communities will be targeted through conference presentations.

Public audiences will be engaged through the production of plain-language summaries. These will be made available through a dedicated study website (http://sphsu.mrc.ac.uk/mup) which research participants will be signposted to (via the participant information leaflet). Posters summarising key results will be made available to study sites, so that findings of the research can be seen by both staff and patients. Broader public engagement will be achieved through an active mass media communication strategy, supported by ancillary use of social media when appropriate.

## Data storage

Except as required for analysis, we have ensured that the electronic questionnaire data, consent forms and data linkage forms are stored separately in three secure databases to maintain full confidentiality for these data.

Any paper-based data collected for the EDs study are stored, de-identified, in locked storage areas with strictly

controlled access and handled according to Medical Research Council (MRC) Good Research Practice Guidelines. Consents to participation in the study and to data linkage are stored separately in a dedicated secured area. Similarly, paper consent and data linkage forms are stored separately in a dedicated secured area.

SHCs data contain no personal identifiers, and are stored securely in a locked storage room and, when entered electronically, in a secure database.

## Data management

The original data were imported for data management and the original files backed up unaltered on a network drive only accessible to the project team. Data cleaning, manipulations and analysis were performed in separate coding/syntax files. We will provide well documented (ie, with extensive commenting) syntax to ensure that all processing of data and results are reproducible.

Ethics review determined study integrity to be dependent on retaining the anonymity of study participants and areas, including comparator sites. The protected period will apply from the start of the study to 1st September 2030.

## Storage and use of data after the end of the study

Long term storage of data is at Iron Mountain, a commercial data storage facility. All data will be held in secure conditions with references for the contents of the sealed boxes held on a database for paper-held data. For electronic data secure electronic archives are used. Access to the data is controlled and permission from the data controller for the study is required to access long-term archived data. The electronic data will be archived at Glasgow University and destroyed by 1 September 2030 (10 years after the end of the study grant) as stated in Research and Development applications to research sites.

Strict data protection policies will be followed as outlined in the University of Glasgow's data protection policy.[82] The data will be worked from and stored on a secure protected server (only accessible to the project team). On completion of the project, the data will then be archived in line with University of Glasgow University guidance on data archiving and the MRC's 'Personal Information in Medical Research' guidance document.

**Author affiliations**
[1]MRC/CSO Social & Public Health Sciences Unit, University of Glasgow School of Life Sciences, Glasgow, UK
[2]Scottish Public Health Observatory, NHS Health Scotland, Glasgow, UK
[3]Addictions, King's College London, London, UK
[4]Institute of Psychiatry, Kings College London, London, UK
[5]School of Health Sciences, Institute for Social Marketing, Stirling, UK
[6]ISD Scotland, Edinburgh, UK
[7]Health Economics Research Unit, University of Aberdeen, Aberdeen, UK
[8]Faculty of Health Sciences Institute for Clinical and Applied Health Research (ICAHR), University of Hull, Hull, UK
[9]Institute for Social Marketing, University of Stirling and the Open University, Stirling, UK
[10]Institute for Social Marketing, University of Stirling, Stirling, UK
[11]Australian Health Policy Collaboration, Victoria University, Victoria, Australia

**Acknowledgements** PD is supported by South London and Maudsley (SLaM) NHS Foundation Trust and by the National Institute for Health Research (NIHR) Biomedical Research Centre (BRC) for Mental Health at King's College London and SLaM. CD is supported by the NIHR Collaborations for Leadership in Applied Health Research South London at King's College Hospital Foundation Trust, the NIHR BRC for Mental Health at King's College London and SLaM, and is in receipt of an NIHR Senior Investigator Award. TP is supported by the NIHR Clinical Research Network for Yorkshire and The Humber. The views expressed here are those of the authors and do not necessarily reflect the views of the Department of Health and Social Care or NIHR. The iPad application (app) used for data collection in Emergency Departments was developed by Codeface Ltd. The authors would like to thank NHS colleagues at all the research sites, who cannot be named for confidentiality reasons, for their assistance with this work. The authors also wish to thank Julie Breslin and Kenneth Crawford of Addaction, who contributed a non-statutory and service user perspective to the Study Steering Committee. In addition, we are particularly grateful to the Population Health Research Facility, and its predecessor, the Social and Public Health Sciences Surveys Unit at the University of Glasgow for their help with non-salary project costing, study training set up, study documents design, study operations set up, data entry and data management.

**Contributors** SVK, LB and CB conceived the study. SVK, CB, CD, DE, LG, SH, AL, GM, MS, LB and AHL all contributed to the overall study design and grant application. LB provided initial strategic leadership for the study, followed by AHL. TP and PD provided additional input on the design and conduct of study component 1 (Emergency Department component). AHL provided statistical expertise for designing study components 1 and 2. AF provided additional input on the design and conduct of study component 3 (Qualitative component). AM and RF led the study day to day, contributed to parts of the methodology used, helped acquire and analyse the data. SVK wrote the first draft of the protocol, which was substantially revised and finalised by AM. All authors made substantial contributions to study development, critically revised the paper and approved the final manuscript.

**Funding** This study was funded by the NIHR Public Health Research Programme (11/3005/40). The Social and Public Health Sciences Unit is funded by the Medical Research Council (MC_UU_12017/13 & MC_UU_12017/15) and Scottish Government Chief Scientist Office (SPHSU13 & SPHSU15). SVK acknowledges funding from a NRS Senior Clinical Fellowship (SCAF/15/02).

**Disclaimer** The views expressed are those of the authors and not necessarily those of the NHS, the NIHR or the Department of Health and Social Care.

**Competing interests** CB, GM and LG are members of the Scottish-Government funded Monitoring and Evaluating Scotland's Alcohol Strategy (MESAS) evaluation.

**Patient consent for publication** Obtained.

**Ethics approval** Ethical approval has been obtained from the NHS through the Scotland A Research Ethics Committee for study components 1 and 2, REC references are 12/SS/0120 and 12/SS/0121. The ethics approval for component 3 was obtained in 2012 from the University of Stirling Management School Ethics Committee (Application numbers 32 and 33).

**Provenance and peer review** Not commissioned; externally peer reviewed.

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
