## [Reviewer comments · BMJ Open]

ARTICLE DETAILS

TITLE (PROVISIONAL)	Evaluating possible intended and unintended consequences of the implementation of alcohol minimum unit pricing (MUP) in Scotland: a natural experiment protocol
AUTHORS	Katikireddi, Srinivasa; Beeston, Clare; Millard, Andrew; Forsyth, Ross; Deluca, Paolo; Drummond, Colin; Eadie, Douglas; Graham, Lesley; Hilton, Shona; Ludbrook, Anne; McCartney, Gerry; Phillips, Thomas; Stead, Martine; Ford, Allison; Bond, Lyndal; Leyland, Alastair

VERSION 1 - REVIEW

REVIEWER	Albert Farre University of Dundee, UK
REVIEW RETURNED	11-Jan-2019

GENERAL COMMENTS	This is an interesting and important study which explores the impact of a minimum alcohol unit pricing intervention in Scotland, in terms of intended and unintended consequences. The paper is well written, well referenced and easy to follow. My only concern would be that, at times, it is difficult to make sense of what elements of the work have already been conducted, which are ongoing and which are planned. I would suggest to perhaps revise the writing to try and make this as clear as possible, adding perhaps more sentences explicitly stating the status of given work components, and avoid the confusion that sometimes arises with a mixture of past and future tenses – for example, whilst it is clear from the text that baseline data collection for component 1 was completed, it would seem as though the 3rd wave has been completed too based on the use of past tense, but 9 months post implementation would be by February 2019 if I understand correctly. This can be confusing as the whole data collection section is written in past tense, but the 'Ethical Arrangements' section then states that "researchers WILL approach and interview participants...". Another example is on page 13 (line 20-21) where the authors state that "focus groups and interviews WERE carried out..." and then indicate the dates, which is great – however, on the next paragraph they go on to say that "participants WILL BE recruited..." (lines 32-34). So I think it would be helpful to just try and add some clarity in terms of what has been completed, what is ongoing and what is planned. I would also suggest to add more detail to the methods for Component 3, which seems a bit too brief as it stands with some key aspects missing (e.g. sampling strategy – I particularly missed an elaboration on the rationale for using different approaches for different stakeholder groups and for employing independent researchers for recruitment; specific data analysis methods
---

	planned and any validation strategies). Tied to this, it would be good to perhaps follow an appropriate checklist for this (such as COREQ) as STROBE would not really be suitable for the reporting of this component. Finally, I would also be interested whether the research team has any plans for data integration across components in terms of seeking to produce an overall, integrated interpretation of their results across components. If so, it would be very interesting to report such data analysis plans in the protocol and any particular approaches and/or frameworks they might plan to use to support this.
--	---

REVIEWER	Samantha Cukier Ottawa Hospital Research Institute, Ontario, Canada
REVIEW RETURNED	16-Feb-2019

GENERAL COMMENTS	This is a protocol of a 3-part study: repeated cross-sectional audit of possible unintended consequences of MUP among members of vulnerable groups (ED study); impact of MUP on alcohol-related behaviours by patients in sexual health clinics; qualitative study to understand narratively the effects of MUP among members of vulnerable groups and policy makers who were part of implementation. The aim of this study is to investigate the impacts of MUP on acute and chronic health conditions and identify the extent to which unintended consequences occur. The results of each branch of this study will contribute to a greater understanding of the policy process in implementing MUP and the consequences – both intended and unintended - of this policy. The protocol is well written and clearly elucidates each of the three studies, provides a solid background with excellent rationale for the studies. The methods, analysis and plans for dissemination are well written and sufficient detail is included in each. The parts of the study already undertaken were clearly reported and the use of the STROBE statement is complete. I look forward to reading the completed study and to see how the information reported can enhance the evidence around MUP. General: In terms of capturing data on vulnerable groups and unintended consequences: the SHC study serves to capture youth and possibly high-risk users. The qualitative study aims to capture individuals living in “deprived” neighborhoods. Will you capture those who are in unstable housing? I am not clear on their inclusion. If they are not included, will you be missing an important high-risk group – and the potential unintended consequences among them? Can you please clarify? Specifics (page numbers referenced here are the stamped BMJ numbers, not the original author pages): Page 6, line 22: “It is also worth noting that reference pricing was originally introduced primarily for its revenue raising potential and not its potential public health benefits.” Can you identify the source for this? The development of reference pricing in different Canadian provinces developed differently and so this may be
---

	steering readers in the wrong direction. Please be more specific in your assertion here. Page 7, line 47: constraints Page 7: Can you provide a rationale for using the FAST? Page 7, line 56: when you say “accidents” do you mean car crashes / collisions? Other injuries? Can you replace “accidents” with another word that is more specific. Accident means that it could not have been prevented. Page 14: Lines 28-37: you go between past tense and future tense (“Group participants will be recruited...”) identifying groups you did conduct interviews with and then others you will conduct interviews with. Can you please clarify which have been completed and which are yet to be complete?
--	--

VERSION 1 – AUTHOR RESPONSE

Reviewers' Comments to Author:

Reviewer: 1

Reviewer Name: Albert Farre

Institution and Country: University of Dundee, UK

Please state any competing interests or state 'None declared': None declared

This is an interesting and important study which explores the impact of a minimum alcohol unit pricing intervention in Scotland, in terms of intended and unintended consequences.

The paper is well written, well referenced and easy to follow. My only concern would be that, at times, it is difficult to make sense of what elements of the work have already been conducted, which are ongoing and which are planned. I would suggest to perhaps revise the writing to try and make this as clear as possible, adding perhaps more sentences explicitly stating the status of given work components, and avoid the confusion that sometimes arises with a mixture of past and future tenses – for example, whilst it is clear from the text that baseline data collection for component 1 was completed, it would seem as though the 3rd wave has been completed too based on the use of past tense, but 9 months post implementation would be by February 2019 if I understand correctly. This can be confusing as the whole data collection section is written in past tense, but the 'Ethical Arrangements' section then states that “researchers WILL approach and interview participants...”. Another example is on page 13 (line 20-21) where the authors state that “focus groups and interviews WERE carried out...” and then indicate the dates, which is great – however, on the next paragraph they go on to say that “participants WILL BE recruited...” (lines 32-34). So I think it would be helpful to just try and add some clarity in terms of what has been completed, what is ongoing and what is planned.

- The tense has been amended to the past for data collection statements at a number of points within the document. Tense remains future for future actions, eg analysis, and the present tense or the infinitive is used where appropriate, eg when referring to the study and the study aims.

I would also suggest to add more detail to the methods for Component 3, which seems a bit too brief as it stands with some key aspects missing (e.g. sampling strategy – I particularly missed an elaboration on the rationale for using different approaches for different stakeholder groups and for employing independent researchers for recruitment; specific data analysis methods planned and any validation strategies). Tied to this, it would be good to perhaps follow an appropriate checklist for this (such as COREQ) as STROBE would not really be suitable for the reporting of this component.

- The qualitative component has been revised by the relevant authors in accordance with the COREQ checklist. These extensive changes are shown using tracked changes in the component 3 section.

Finally, I would also be interested whether the research team has any plans for data integration across components in terms of seeking to produce an overall, integrated interpretation of their results across components. If so, it would be very interesting to report such data analysis plans in the protocol and any particular approaches and/or frameworks they might plan to use to support this.

- We would refer you to Table 1 about how we expect different components to be triangulated to answer each of the three RAs. A statement has now also been added to the Dissemination plan section (page 15) as follows:

“We will produce an overall theoretical interpretation utilising all three components in the final study report.”

Reviewer: 2

Reviewer Name: Samantha Cukier

Institution and Country: Ottawa Hospital Research Institute, Ontario, Canada

Please state any competing interests or state ‘None declared’: None declared

This is a protocol of a 3-part study: repeated cross-sectional audit of possible unintended consequences of MUP among members of vulnerable groups (ED study); impact of MUP on alcohol-related behaviours by patients in sexual health clinics; qualitative study to understand narratively the effects of MUP among members of vulnerable groups and policy makers who were part of implementation. The aim of this study is to investigate the impacts of MUP on acute and chronic health conditions and identify the extent to which unintended consequences occur.

The results of each branch of this study will contribute to a greater understanding of the policy process in implementing MUP and the consequences – both intended and unintended - of this policy. The protocol is well written and clearly elucidates each of the three studies, provides a solid background with excellent rationale for the studies. The methods, analysis and plans for dissemination are well written and sufficient detail is included in each. The parts of the study already undertaken were clearly reported and the use of the STROBE statement is complete. I look forward to reading the completed study and to see how the information reported can enhance the evidence around MUP.

General:

In terms of capturing data on vulnerable groups and unintended consequences: the SHC study serves to capture youth and possibly high-risk users. The qualitative study aims to capture individuals living in “deprived” neighborhoods. Will you capture those who are in unstable housing? I am not clear on their inclusion. If they are not included, will you be missing an important high-risk group – and the potential unintended consequences among them? Can you please clarify?

See page 7:

- ‘Another separately funded study is focusing on the impacts on people with alcohol dependence, including high rates of homelessness specifically. We therefore do not target this group for our qualitative work.’
- The Emergency Department component includes a question on housing status. We have given more detail about this in the data collection section within the component 1 methods (see page 9)

‘The interview collected details on the following:

- basic demographic information (age; gender; ethnicity; marital, employment and housing status (including unstable housing status such as no fixed abode and hostel accommodation); postcode sector);’

Specifics (page numbers referenced here are the stamped BMJ numbers, not the original author pages):

Page 6, line 22: “It is also worth noting that reference pricing was originally introduced primarily for its revenue raising potential and not its potential public health benefits.” Can you identify the source for this? The development of reference pricing in different Canadian provinces developed differently and so this may be steering readers in the wrong direction. Please be more specific in your assertion here.

- Since this point is not essential to the paper, we have deleted this sentence and amended the following one (page 5).
~~‘It is also worth noting that reference pricing was originally introduced primarily for its revenue raising potential and not its potential public health benefits. Important benefits of reference pricing in Canada have nevertheless been demonstrated, with reductions in alcohol consumption, alcohol-attributable hospital admissions and crime observed.’^{1-6’}~~

Page 7, line 47: constraints

- Thankyou – typo in the word ‘constraints’ has been corrected (page 6)

Page 7: Can you provide a rationale for using the FAST?

- See page 6: ‘The use of FAST to identify alcohol misuse (i.e. hazardous, harmful and dependent drinking) has been supported by national guidelines for the prevention of harmful drinking⁷³ (NICE, 2010). Derived from the AUDIT⁷⁴ the four item FAST was specifically developed and validated within ED as it is quick to administer and less disruptive in the busy ED setting.^{72 75} The FAST has been found to be the most sensitive and accurate short alcohol screening tool commonly administered to identify alcohol misuse in adults attending ED ^{76 77} and has been used in a number of clinical studies.^{78 79’}
- Page 8: ‘The FAST is used identify alcohol misuse in the ED. The component focuses on answering RA1, namely:’
 Page 7, line 56: when you say “accidents” do you mean car crashes / collisions? Other injuries? Can you replace “accidents” with another word that is more specific. Accident means that it could not have been prevented.

The word ‘accident’ has been deleted or if appropriate replaced with other more appropriate language at various points in the document in line with current thinking that all injuries are preventable:

Page 4: ‘Alcohol is associated with over 200 medical conditions including an increased risk of liver disease, heart disease, unintended pregnancy, sexually transmitted infections, some cancers and ~~accidental~~ some injuries.9-14’

Page 7 ‘Hazardous drinkers are expected to be more likely than harmful or dependent drinkers to present with acute harms (such as ~~accidents~~ injuries and assaults), while harmful/dependent drinkers are likely to present with both acute conditions and acute consequences of chronic conditions (such as pancreatitis and gastritis).’

- 'attendance details (date and time, triage category, time and date of accident/injury incident),'
...
- 'self-reported reason for attendance, such as assault, accident collapse etc; '

Page 14: Lines 28-37: you go between past tense and future tense ("Group participants will be recruited...") identifying groups you did conduct interviews with and then others you will conduct interviews with. Can you please clarify which have been completed and which are yet to be complete?

- The amendments to component three are quite extensive and are shown in the revised document. They do now use the past tense for data collection activities as those are now complete (pages13-15).

VERSION 2 – REVIEW

REVIEWER	Albert Farre University of Dundee, UK
REVIEW RETURNED	12-Apr-2019

GENERAL COMMENTS	Thank you for your responses and clarifications, and for incorporating the reviewer's suggestions into the revised manuscript. I think the changes have improved clarity for readers on some aspects of this study which in the previous version of the manuscript were perhaps less clear. However, as I noted in my previous review, this was already a well written and easy to follow protocol paper. So congratulations to the authorship group for improving this further. This is a very important research topic and I look forward to seeing the findings of this study published in the future.
---

REVIEWER	Samantha Cukier Ottawa Hospital Research Institute, Canada
REVIEW RETURNED	16-Apr-2019

GENERAL COMMENTS	The authors have addressed all comments thoroughly. This is a well thought out protocol that will certainly add to the literature on MUP.
---